

# Constraining urban biogenic $CO_2$ fluxes: Composition, seasonality and drivers from radiocarbon and inventory analysis

Pingyang Li[1,2], Boji Lin[1,2,3], Zhihua Zhou[4,5,*], Jing Li[1,2], Zhineng Cheng[1,2], Jun Li[1,2], Sanyuan Zhu[1,2], Shizhen Zhao[1,2], Guangcai Zhong[1,2], Gan Zhang[1,2,*]

[1] State Key Laboratory of Advanced Environmental Technology, Guangzhou Institute of Geochemistry, Chinese Academy of Sciences, Guangzhou 510640, People's Republic of China
[2] Guangdong Key Laboratory of Environmental Protection and Resources Utilization, and Joint Laboratory of the Guangdong-Hong Kong-Macao Greater Bay Area for the Environment, Guangzhou Institute of Geochemistry, Chinese Academy of Sciences, Guangzhou 510640, People's Republic of China
[3] University of Chinese Academy of Sciences, Beijing 100049, People's Republic of China
[4] Shenzhen Ecological and Environmental Monitoring Center of Guangdong Province, Shenzhen 518049, People's Republic of China
[5] Guangdong Greater Bay Area, Change and Comprehensive Treatment of Regional Ecology and Environment, National Observation and Research Station, Shenzhen 518049, People's Republic of China

*Correspondence to*: Gan Zhang (zhanggan@gig.ac.cn), Zhihua Zhou (zhou@zhihua.org)

**Abstract.** Urban areas play a pivotal role in achieving net-zero emissions to limit global warming to 1.5°C, given their high carbon footprint and mitigation potential. Accurate quantification of urban $CO_2$ sources is essential for effective carbon budgeting and targeted climate action. While fossil fuel $CO_2$ ($CO_{2ff}$) emissions are extensively studied, biogenic $CO_2$ ($CO_{2bio}$) dynamics remain poorly constrained. Here, we separate fossil and biogenic contributions to $CO_2$ enhancements above background using $\Delta^{14}C$ and $CO_2$ measurements in Shenzhen, a humid subtropical Chinese megacity potentially subject to substantial biomass burning influence. We calculate human/livestock metabolic emissions ($CO_{2HLM}$) at 9.32 Mt/6.22 kt per year from population/livestock data and respiratory/excretory rates, and estimate biomass burning emissions ($CO_{2BB}$) at 5.05 Mt/yr using an inventory encompassing both open and domestic combustion. The residual $CO_{2bio}$ component is attributed to the terrestrial biosphere ($CO_{2bio}'$). Integrating $\Delta^{14}C$ with multi-source data reveals annual $CO_{2bio}$ contributions relative to fossil fluxes: $CO_{2HLM}$ (17.8 ± 3.1%), $CO_{2BB}$ (9.2 ± 1.5%), and $CO_{2bio}'$ (73.0 ± 3.5%). Key findings demonstrate the terrestrial biosphere component acts as a year-round net carbon sink with significant seasonality (11.5 ppm amplitude; ~1.5 times the annual mean $CO_{2ff}$ concentration), driven primarily by atmospheric temperature (1-2 months lag; $r = -0.80$, $p = 0.01$) rather than precipitation. This study establishes human metabolic emissions as the dominant biogenic $CO_2$ source (17.8% vs. 9.2% from biomass burning) in megacities, yet shows that concurrent biospheric sequestration can offset 63% of fossil emissions during growing seasons, advancing understanding of urban carbon budgets.



## 1 Introduction

Accelerating global urbanization underscores the critical role of cities, as core hubs of anthropogenic carbon emissions, in achieving the 1.5°C climate target (Duren and Miller, 2012; Seto et al., 2021). Given their high carbon footprints and significant mitigation potential, accurate quantification of urban $CO_2$ emission sources is essential for effective carbon
budgeting and climate action. However, current understanding remains heavily skewed toward fossil fuel $CO_2$ ($CO_{2ff}$) (Levin et al., 2008; Turnbull et al., 2015; Newman et al., 2016; Wang et al., 2022), while the dynamics of biogenic $CO_2$ ($CO_{2bio}$)— encompassing human/livestock metabolism, biomass burning, and terrestrial biosphere fluxes—are poorly constrained. This knowledge gap is particularly critical in humid subtropical megacities, where substantial but inadequately quantified biomass burning influence may severely hinder comprehensive urban carbon budget assessments and the design of targeted
mitigation strategies.

Although monitoring and modelling of urban $CO_{2ff}$ emissions have matured (e.g., inventory-based approaches using economic statistics, atmospheric $CO_2$ concentration inversions), research on $CO_{2bio}$ components remains nascent. Existing studies predominantly focus on temperate cities (e.g., Paris, Los Angeles, Beijing, Xi'an) (Lopez et al., 2013; Miller et al., 2020; Zhou et al., 2020) or single emission sources (e.g., human respiration, livestock respiration) (Cai et al., 2022; Wang et
al., 2024b), lacking integrated analysis of biogenic $CO_2$ subcomponents (e.g., human/livestock metabolic emissions, open/domestic biomass combustion, ecosystem respiration and uptake). Significant gaps persist, especially in humid subtropical megacities, regarding the seasonal dynamics, driving mechanisms, and interactions with fossil emissions of $CO_{2bio}$. This may cause underestimation of urban carbon sink potential and creates blind spots in carbon neutrality management strategies.

Addressing these limitations necessitates the development of multi-dimensional observation techniques and multi-source data fusion. Radiocarbon isotope ($\Delta^{14}C$) analysis serves as a unique tracer for distinguishing fossil from biogenic $CO_2$ (Levin et al., 2003; Turnbull et al., 2006), because $CO_{2ff}$ contains no detectable $^{14}C$ due to complete radioactive decay over geological timescales. However, comprehensive resolution of $CO_{2bio}$ components demands integration of $\Delta^{14}C$ with $CO_2$ concentration monitoring, population/livestock metabolic flux calculations (Cai et al., 2022; Miller et al., 2020; Wang et al.,
2024b), and biomass burning emission inventories (Randerson et al., 2017; Van Der Werf et al., 2017; Edgar, 2024). By coupling $\Delta^{14}C$ tracing with respiratory/excretory flux models and biomass burning inventories, a systematic separation of metabolic ($CO_{2HLM}$), biomass burning ($CO_{2BB}$), and net terrestrial biosphere fluxes ($CO_{2bio}'$) becomes feasible, thereby revealing the precise contribution of biogenic $CO_2$ to urban carbon cycling.

To bridge these gaps, this study focuses on Shenzhen, a representative humid subtropical megacity in China. Using coupled
$\Delta^{14}C$ and $CO_2$ observations (over more than a year at five sites) as the methodological cornerstone, we quantitatively separate $CO_{2ff}$ and $CO_{2bio}$. By further integrating multi-source datasets, we determine the absolute concentrations and relative contributions of three major biogenic $CO_2$ components: $CO_{2HLM}$, $CO_{2BB}$, and $CO_{2bio}'$. We then analyse the seasonal dynamics,





driving factors, and carbon sink effects of $CO_{2bio}'$, clarifying its critical role in urban carbon budgets. This work provides a scientific basis for designing carbon-neutral pathways—particularly nature-based solutions—in global megacities.

## 2 Data and methods

### 2.1 Study area and sample collection

We conducted atmospheric sampling of $CO_2$ and its radiocarbon isotope ($\Delta^{14}C$) in Shenzhen (22.45°N−22.87°N, 113.77°E−114.62°E), a rapidly developing high-tech megacity in South China (Fig. 1ab). Geographically, Shenzhen lies north of Hong Kong and has a monsoon influenced humid subtropical climate. As China's first "special economic zone" established in 1978, Shenzhen has grown into a global economic hub, with a population of 17.66 million and a GDP of 3.248 trillion yuan in 2022 (Smbs, 2024), surpassing the economic output of over half of China's provinces. As one of the core cities in the Guangdong–Hong Kong–Macao Greater Bay Area, the world's largest urban agglomeration, Shenzhen leads in low-carbon development, driven primarily by its computer, communication, and electronic equipment manufacturing sectors.

Both bottom-up and top-down studies could add to our understanding of Shenzhen's substantial urban $CO_2$ fluxes. According to the near-real-time Global Gridded Daily $CO_2$ Emissions Dataset (GRACED) (Liu et al., 2020), Shenzhen's total $CO_{2ff}$ emissions in 2022 were 55.0 $TgCO_2$, primarily from power (55%) and industry (25%); while the bottom-up Multi-resolution Emission Inventory for China (MEIC) (Meic, 2023) reported 44.7 $TgCO_2$ in 2020, primarily from industry (55%) and transportation (30%). In contrast, the Carnegie Ames Stanford Approach Global Fire Emissions Database Version 4 (CASA-GFED4s) dataset (Randerson et al., 2017; Van Der Werf et al., 2017) estimates a Net Ecosystem Exchange (NEE) flux of −0.19 $TgCO_2$ in 2015, highlighting the city's significant carbon sink potential. Shenzhen's land use is dominated by terrestrial ecosystems (64%), including urban areas (49%), forests (44%), and wetlands (4%) (Ssb, 2024), with marine ecosystems covering the remaining 36%. To mitigate urban expansion and preserve natural landscapes, Shenzhen implemented a "Basic Ecological Control Line" in 2005, effectively curbing forest loss (Yu et al., 2016) and promoting sustainable development.

From April 2022 to April 2023, we collected air samples once or twice weekly between 13:00 and 17:00 local standard time at five strategically selected sites (Fig. 1c): 30-m towers in parks in western areas near the Pearl River estuary (SZ1), in central suburban areas (SZ4), and in eastern suburban areas (SZ5), as well as 10−12 m masts on building rooftop corners in southern downtown (SZ2, 200 m above ground level (m a.g.l.)) and near the city's northeastern boundary (SZ3, 110 m a.g.l.). This sampling design provided uniform spatial coverage across Shenzhen and was assumed statistically representative for city-wide analysis (Fig. 1d). Air samples were collected by drawing filtered air through evacuated, pre-purged 6 L SilcoCan canisters and 3 L borosilicate glass flasks connected in series. Using 12 V micro diaphragm gas pumps, air was drawn at a flow rate of 6 L $min^{-1}$ and pressurized to 25−30 psi. Each sampling session lasted approximately 30 min.





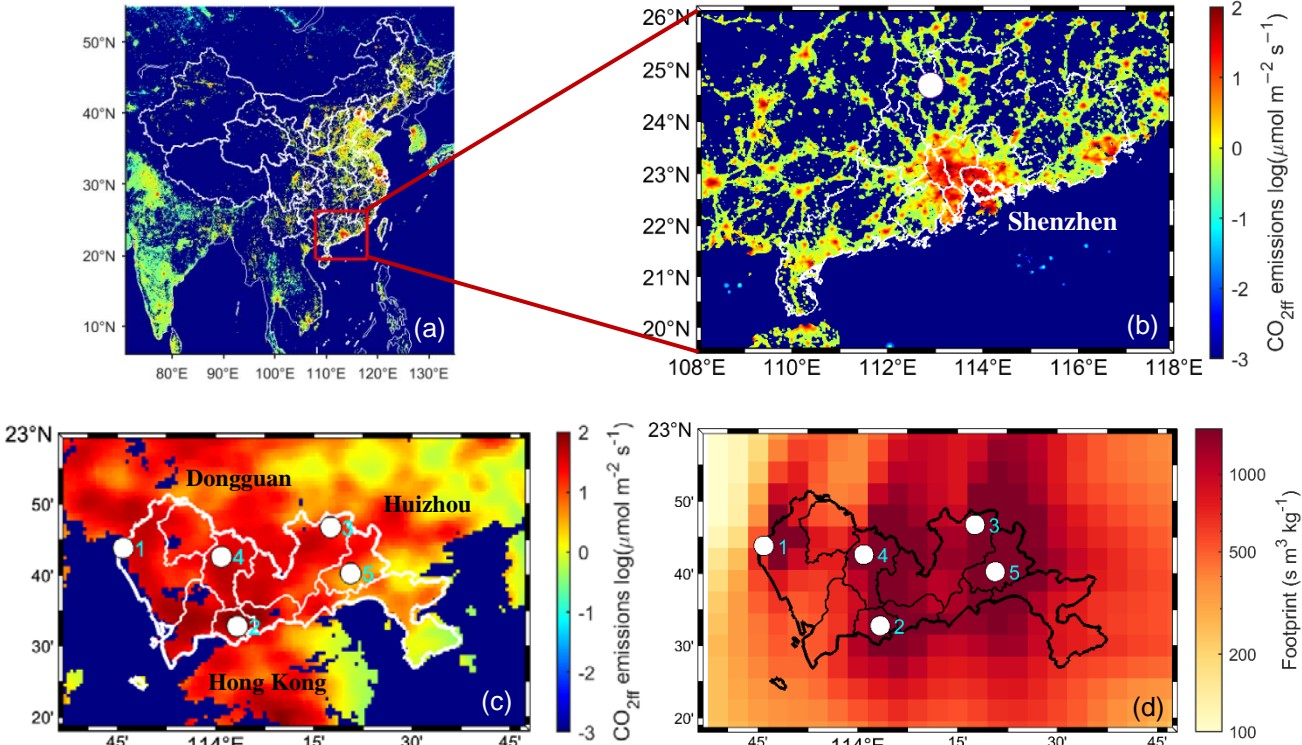

**Figure 1: (a-c) Geographical distribution of five Shenzhen sites (SZ1–SZ5) and (d) their surface flux sensitivity (FLEXPART footprint) during the sampling period (April 2022 – April 2023). White lines in (a) represent continental and province boundaries, which were obtained from Natural Earth (https://www.naturalearthdata.com/, last accessed on 9 June 2024). In (b), the circle represents the location of Nanling background site. The borders of Guangdong province and the nine Pearl River Delta (PRD) cities were marked with white lines, with Shenzhen marked with a bold white line. Shading in (a-c) indicates $CO_{2ff}$ emissions at a spatial resolution of 1km × 1km from the Open-source Data Inventory for Anthropogenic $CO_2$ (ODIAC) (Oda and Maksyutov, 2024), which share the same color bar. In (d), the domain of the footprints (which go back 30 days) is at a spatial resolution of 0.05° × 0.05°.**

## 2.2 Measurement of CO₂

We extracted air from the canisters to determine $CO_2$ mole fractions using a modified gas chromatograph with flame ionization detection (GC-FID) system (Agilent 7890B, Agilent Technologies Inc., USA). Samples were introduced by flushing a 5 mL sample loop, after which $CO_2$ was separated through a HayeSep Q packed column. The isolated $CO_2$ was then converted to $CH_4$ via a nickel catalyst furnace at 375°C and subsequently quantified by FID. Calibration of $CO_2$ mixing ratios was conducted using three reference standards, obtained from the National Center for Reference Materials Research, China Institute of Metrology. These standards are traceable to the X2019 calibration scale maintained by the Central Calibration Laboratory of the World Meteorological Organization. The precision of $CO_2$ measurements was better than 0.2 ppm (Zhou et al., 2024).



## 2.3 Measurement of $\Delta^{14}C$

We extracted air from the flasks for $\Delta^{14}C$ measurement using established cryogenic techniques. This involved using a cold trap with a dry ice and ethanol slurry (about $-70^\circ C$) to remove water, followed by a liquid nitrogen cold trap ($-196^\circ C$) to condense $CO_2$ (Xu et al., 2007). The purified $CO_2$ samples were then graphitized using the hydrogen reduction method and analyzed for $\Delta^{14}C$ content with an NEC 0.5MV 1.5SDH-2 accelerator mass spectrometer (AMS, National Electrostatics Corporation, USA) (Zhu et al., 2015). Each measurement wheel typically comprises 13 oxalic acid II as primary standards,

13 IAEA-C7 as secondary standards, 13 $p$-phthalic acid as solid process blanks, 6 $^{14}C$-free $CO_2$ in synthetic air from a cylinder as gas process blanks, and some authentic air samples. Results are presented as $\Delta^{14}C$, which is the per mill (‰) deviation from the absolute radiocarbon reference standard, corrected by fractionation and decay (Stuiver and Polach, 1977):

$$\Delta^{14}C = [\frac{A_{SN}}{A_{ON}e^{\lambda(y-1950)}} - 1] \times 1000‰ \quad (1)$$

where $A_{SN}$ and $A_{ON}$ denote the $^{14}C/^{12}C$ ratios of sample and oxalic acid II reference standard, respectively. $\lambda$ represents the

decay constant of $^{14}C$ with a value of 1/8267 $yr^{-1}$. y is the year of sampling, and 1950 is the reference year for "modern".

We analyzed 22 paired air samples to evaluate the quality control and quality assurance of the entire sampling and laboratory analysis process, including sampling, extraction, graphitization and AMS measurement. The AMS measurement uncertainty and average deviation were $2.2 \pm 0.4$‰ and $1.9 \pm 2.4$‰, respectively. The one-sigma measurement uncertainty was thus 2.4‰ for $\Delta^{14}C$ based on repeat measurements of 22 pairs of parallel air samples (Fig. S1).

## 2.4 Background

To quantify $CO_2$ enhancements in Shenzhen, we would like to obtain an upwind urban background. However, we found that neither any single observational site (e.g., winter upwind background SZ3 or eastern suburban background SZ5) nor the monthly Keeling plot background (top-right corner value) (Zhou et al., 2024; Li et al., 2025) could reflect the regular monthly variations of $CO_{2bio}$ concentrations. The available background options (Fig. 2) were either: (1) the overall

background from urban Keeling plots for all months (Kpa; giving $421.4 \pm 0.1$ ppm and $0.1 \pm 0.8$ ‰ for $CO_2$ and $\Delta^{14}C$, respectively, by averaging the highest 2 % top-right corner values), or (2) the nearest regional background monthly means from Nanling (NL, 1700 m above sea level (m a.s.l.)).

Previous studies in megacities often calculated atmospheric $CO_2$ concentrations relative to the regional background (Table S1) due to locally emitted $CO_2$ (Newman et al., 2016; Miller et al., 2020; Wang et al., 2022; Zazzeri et al., 2023). In this

study, to keep the monthly variations of the background, we defined representative background $CO_2$ and $\Delta^{14}C$ levels using measurements from the nearest regional background NL site. The station samples were relatively clean, well-mixed free tropospheric air. Background $CO_2$ concentrations were derived from in-situ measurements using a G2301 Gas Concentration Analyzer (Picarro Inc., USA) based on the fourth-generation wavelength scanning cavity ring-down spectroscopy. These



data were filtered by excluding measurements taken during wind speeds below 2 m s⁻¹, followed by a robust extraction of
baseline signals (Fig. 2a). A curve from Thoning et al. (1989) was then fitted to the baseline, applying a threshold of less
than two sigma deviations (Zhang et al., 2022). Background $\Delta^{14}C$ values were directly measured (Fig. 2b). Quality control
measures excluded obvious low $\Delta^{14}C$ outliers, corresponding to air with extremely high $CO_{2ff}$ values. We obtained a total of
251 measurements for $\Delta^{14}C$ and $CO_2$ at SZ1 ($n = 48$), SZ2 ($n = 52$), SZ3 ($n = 52$), SZ4 ($n = 48$), and SZ5 ($n = 51$).

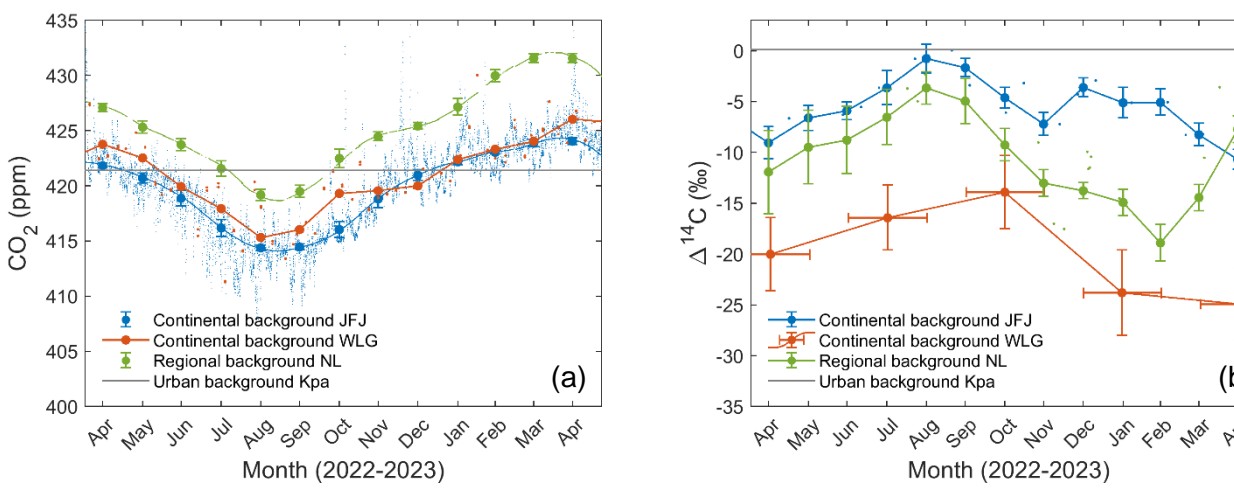

**Figure 2: Background curves and data used to construct them at JFJ, WLG, NL, and Kpa for (a) $CO_2$ and (b) $\Delta^{14}C$. JFJ
(Jungfraujoch, 3580 m a.s.l.) $CO_2$ and $\Delta^{14}C$ measurements are sourced from the Integrated Carbon Observation System (ICOS)
(Emmenegger et al., 2024a, b). WLG (Waliguan, 3890 m a.s.l.) $CO_2$ measurements are from the National Oceanic and
Atmospheric Administration (NOAA) (Lan et al., 2024), while $\Delta^{14}C$ is extrapolated from Liu et al. (2024) using a 4.92‰ yr⁻¹
decline rate, which is interpolated between 2015 (Niu et al., 2016) and 2021–2022 (Liu et al., 2024). NL (Nanling, 1700 m a.s.l.) $CO_2$**
**and $\Delta^{14}C$ measurements are from this study. Kpa $CO_2$ and $\Delta^{14}C$ measurements are determined by averaging the highest 2 % top-
right corner samples from the five sites in the Keeling plot.**

## 2.5 $CO_{2ff}$ and $CO_{2bio}$ concentration estimation by mass balance equations

Recently added atmospheric $CO_2$ ($CO_{2obs}$) comprises background $CO_2$ ($CO_{2bg}$) and excess $CO_2$ ($CO_{2xs}$). $CO_{2xs}$ is dominated
by contributions from fossil fuel combustion ($CO_{2ff}$), and biogenic sources ($CO_{2bio}$). The corresponding $\Delta^{14}C$ are denoted as
$\Delta_{obs}$ (observed), $\Delta_{bg}$ (background), $\Delta_{ff}$ (−1000 ‰, representing zero ¹⁴C content) and $\Delta_{bio}$, respectively. The governing mass
balance equations for atmospheric $CO_2$ and $\Delta^{14}C$ are defined as follows:

$$CO_{2obs} = CO_{2bg} + CO_{2xs} = CO_{2bg} + CO_{2ff} + CO_{2bio} \qquad (2)$$

$$CO_{2obs}\Delta_{obs} = CO_{2bg}\Delta_{bg} + CO_{2ff}\Delta_{ff} + CO_{2bio}\Delta_{bio} \qquad (3)$$

$$CO_{2ff} = \frac{CO_{2obs}(\Delta_{obs}-\Delta_{bg})}{\Delta_{ff}-\Delta_{bg}} - \frac{CO_{2bio}(\Delta_{bio}-\Delta_{bg})}{\Delta_{ff}-\Delta_{bg}} = \frac{CO_{2obs}(\Delta_{obs}-\Delta_{bg})}{\Delta_{ff}-\Delta_{bg}} - \beta \qquad (4)$$





$$CO_{2bio} = CO_{2obs} - CO_{2bg} - CO_{2ff} \approx CO_{2HLM} + CO_{2BB} + (CO_{2r} - CO_{2p}) \qquad (5)$$

The added $CO_{2ff}$ and $CO_{2bio}$ components were calculated using Eqs. 4 and 5, respectively. Contributions of $CO_2$ and $\Delta^{14}C$ from non-target sources, including air-sea exchange and nuclear facilities, were excluded from our analysis due to documented insignificance (Graven et al., 2018; Li et al., 2025). Specifically, Shenzhen's nuclear power plants (Daya Bay

and Ling'ao pressurized water reactors) emit hydrocarbons as their primary effluent (75–95% of total emissions), with $^{14}CO_2$ releases being orders of magnitude significantly below legal thresholds (Li et al., 2025; Meec, 2021).

The second term in Eq. 4 represents a small correction ($\beta$) to account for the effect of $CO_2$ sources from biospheric exchange with a slightly different $\Delta^{14}C$ value compared to atmospheric $\Delta^{14}C$. This primarily reflects the influence of heterotrophic respiration (Rh) and biomass burning (BB) (Graven et al., 2018; Li et al., 2025), since $\Delta^{14}C$ of autotrophic respiration and

photosynthetic uptake is implicitly assumed to be equal to $\Delta_{bg}$ (Turnbull et al., 2006; Turnbull et al., 2009). $\beta$ was quantified using integrated modeling frameworks developed in Li et al. (2025), with key implementation details provided in the next section. The heterotrophic respiration correction ($\beta_{Rh}$, $-0.09 \pm 0.05$ ppm; range: $-0.32$ to $-0.01$ ppm) was derived from FLEXPART simulations (Pisso et al., 2019) combined with CASA-GFED4s dataset (Randerson et al., 2017; Van Der Werf et al., 2017). The biomass burning corrections ($\beta_{BB}$, maximum $-0.18 \pm 0.12$ ppm; range: $-0.94$ to $-0.01$ ppm) was calculated

from FLEXPART simulations combined with EDGAR2024 inventory (Edgar, 2024). The combined correction ($\beta = \beta_{Rh} + \beta_{BB}$) yielded $-0.27 \pm 0.16$ ppm (range: $-1.07$ to $-0.02$ ppm), which is between corrections in summer ($-0.5 \pm 0.2$ ppm) and winter ($-0.2 \pm 0.1$ ppm) reported by Turnbull et al. (2009). We used simulated corrections for each sample.

## 2.6 Simulation by FLEXPART dispersion model

Model simulations were performed using the FLEXPART (FLEXible PARTicle) dispersion model, version 10.4 (Pisso et al.,

2019), a Lagrangian particle dispersion tool. This model generates source-receptor relationships, often known as "footprints", for atmospheric surface measurements by simulating the transport of air parcels to the sampling sites. The simulations account for advection, random diffusion, and atmospheric turbulence, using predefined time intervals and a set number of hypothetical particles. They were driven by global meteorological fields from the Climate Forecast System (CFSv2) Reanalysis model provided by the National Centers for Environmental Prediction (Saha et al., 2011). We determine the mole

fraction enhancement above the background at a specific time by multiplying the model-generated footprints with $CO_2$ fluxes derived from spatially gridded fluxes and integrating over the domain. The footprints were calculated by releasing 10 000 virtual particles from sampling sites and tracking their backward trajectories over 30 days. Covering the domain at $0.05° \times 0.05°$ resolution, these footprints were combined with $CO_2$ inventory emissions to produce simulated concentrations.

For terrestrial biospheric $CO_2$ simulations, we convolved hourly FLEXPART footprints with terrestrial biospheric $CO_2$

fluxes. We employed monthly Net Ecosystem Exchange (NEE) fluxes derived from the Carnegie Ames Stanford Approach Global Fire Emissions Database Version 4 (CASA-GFED4s) model (Randerson et al., 2017; Van Der Werf et al., 2017).



These monthly fluxes were resolved into hourly emissions by imposing the diurnal cycle from the CASA-GFED3 model (Van Der Werf et al., 2010) onto the nearest CASA-GFED4s monthly fluxes, thus approximating hourly terrestrial biosphere fluxes.

For the heterotrophic respiration correction term ($\beta_{Rh}$), $CO_{2Rh}$ mole fractions were estimated by convolving hourly FLEXPART footprints with heterotrophic respiration fluxes from CASA-GFED4s. These fluxes were temporally downscaled from monthly resolutions by imposing the CASA-GFED3 diurnal cycle. Notably, heterotrophic respiration in CASA-GFED3 was estimated as half of the ecosystem respiration, which is calculated as the difference between NEE and gross ecosystem exchange (GEE) fluxes (i.e., [NEE–GEE]/2). The $\Delta^{14}C$ signature for heterotrophic respiration was wet to

40 ± 35‰, derived by applying a secular decline rate of 5‰ yr$^{-1}$ (Zazzeri et al., 2023) to the 2015 measurement of 75 ± 35‰ (Graven et al., 2018), extrapolated to our study period.

For the biomass burning correction term ($\beta_{BB}$), $CO_{2BB}$ mole fractions were estimated by convolving FLEXPART footprints with biomass burning fluxes from EDGAR2024. This inventory was selected because—unlike alternatives such as CASA-GFED4s which exclusively quantify satellite-observable open burning (wildfires, agricultural residue burning, and

savanna/rangeland fires)—EDGAR2024 comprehensively incorporates both open combustion and anthropogenic domestic emissions (including residential biomass heating and industrial biofuel use). To bound the maximum plausible contribution, we conducted FLEXPART simulations under two key assumptions: (1) $CO_{2bio}$ emissions in EDGAR2024 originate entirely from biomass burning ($CO_{2BB}$ = 100% $CO_{2bio}$), and (2) $\Delta^{14}C$ endmembers represent exclusively multi-year biomass. The $\Delta^{14}CO_2$ signature for multi-year biomass burning (116.2 ± 17.6‰ in 2022) was adopted from Li et al. (2025).

This approach allowed quantification of $\beta_{BB}$ under a conservative, upper-limit biomass burning scenario.

## 3 Results and discussion

### 3.1 $CO_2$, $\Delta^{14}C$, $CO_{2ff}$, and $CO_{2bio}$ concentrations

Figure 3 shows large and seasonally varying concentrations of $CO_2$ and $\Delta^{14}C$, and the corresponding estimates of $CO_{2ff}$ and $CO_{2bio}$ at the five sampling sites across the Shenzhen megacity, along with background values from the nearest regional

background site. During the study period, the mean $CO_2$ concentration was 431.1 ± 5.9 ppm (Fig. 3a), representing an enhancement of 5.8 ± 3.0 ppm relative to the background ($CO_{2xs}$, multisite mean and one-sigma standard deviation). The average $\Delta^{14}C$ value was –27.7 ± 9.9‰ (Fig. 3b), with a depletion of –17.2 ± 8.0‰ ($\Delta\Delta^{14}C$) relative to the background, indicating a significant influence of $^{14}C$-free $CO_2$ emissions, primarily from fossil fuel combustion.

The fossil fuel and biogenic contributions to $CO_{2xs}$, denoted as $CO_{2ff}$ and $CO_{2bio}$, were estimated through a two end-member

mixing analysis. This yielded proportions of 93.9% and 6.1% respectively during winter—though the low $CO_{2bio}$ ratio likely reflects an underestimate due to the inclusion of negative values, implying higher actual biogenic contributions. The average $CO_{2ff}$ over the study period was 7.7 ± 3.7 ppm, with values ranging from –5.0 ppm to 26.4 ppm (Note that some negative,





nonphysical, $CO_{2ff}$ are expected; Fig. 3c). Mean $CO_{2ff}$ concentrations were higher in winter (December – February; $9.9 \pm 0.6$ ppm) compared to summer (May – September; $6.8 \pm 4.3$ ppm), likely attributable to increased atmospheric trapping of

emissions within the shallow boundary layer and elevated emissions, as indicated by GRACED (MEIC) inventories, which show that winter emissions are 22% (25%) higher than summer emissions. $CO_{2ff}$ concentrations in Shenzhen are relatively low compared to other cities, such as Paris and Los Angeles (Table S1). This aligns with Shenzhen's role as a pioneer in leading China's and global low-carbon development.

$CO_{2bio}$ concentrations (Fig. 3d) are calculated as the difference between $CO_{2xs}$ and $CO_{2ff}$ (Eq. 2). It shows an annual mean

value of $-1.8 \pm 4.1$ ppm (mean and standard deviation of monthly means), with values ranging from $-17.2$ ppm to $22.2$ ppm. It also shows a mean wintertime enhancement of $-0.4 \pm 2.0$ ppm and a more pronounced summertime mean enhancement of $-2.6 \pm 5.1$ ppm. Notably, 71.7% of all $CO_{2bio}$ measurements were negative, indicating some degree of net $CO_2$ uptake, especially in summer. This leads to $CO_{2bio}$ concentrations in Shenzhen relatively low compared to other cities (Table S1). They are lower than those in Paris (4.5 to 10.2 ppm in winter 2010) and Los Angeles ($-0.3 \pm 1.0$ ppm in summer 2015), and

comparable with those in London ($-17$ to $-3$ ppm in summer 2020). Noting that the concentration of $CO_{2bio}$ is directly dependent on background selection. The consistent adoption of a regional background enables the comparison of $CO_{2bio}$ concentrations across these cities.

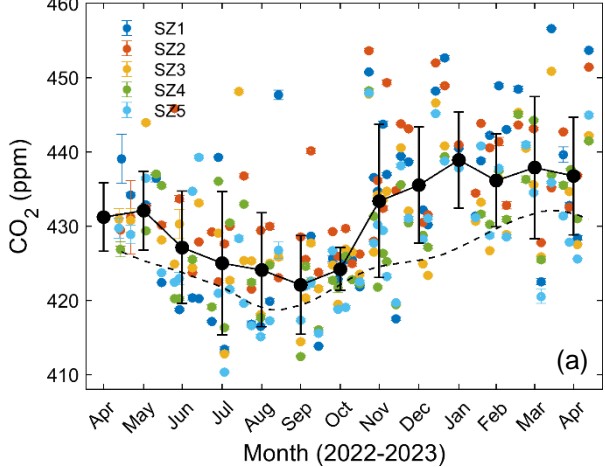

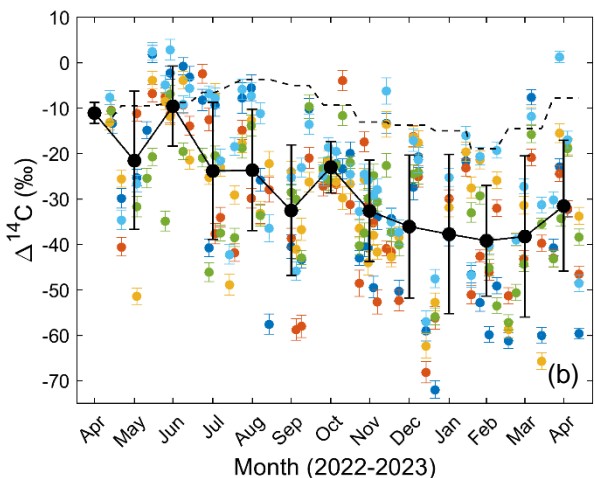



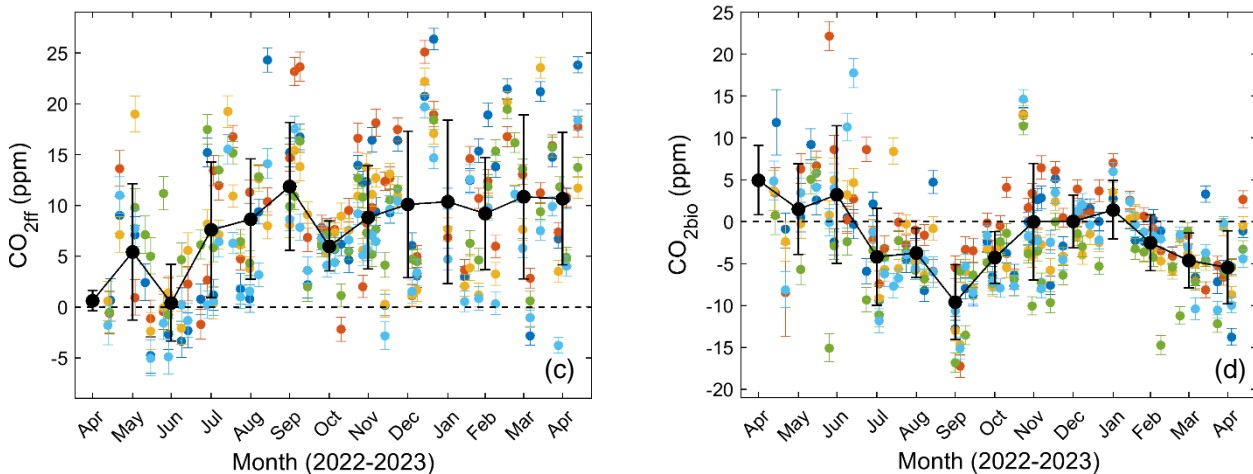

**Figure 3: Time series of (a and b) measured ($CO_2$ and $\Delta^{14}C$) and (c and d) derived quantities ($CO_{2ff}$ and $CO_{2bio}$) from five different sampling sites (SZ1–SZ5). Monthly mean values are overplotted as large black circles with one-sigma standard deviations. Black dashed lines and pluses in (a) and (b) are the NL (Nanling background) curves and data.**

## 3.2 $CO_{2bio}$ components

Urban $CO_{2bio}$ components mainly include contributions from human/livestock metabolism (i.e., respiration and excretion; $CO_{2HLM}$), biomass burning (including open and domestic burning; $CO_{2BB}$), and terrestrial biosphere (i.e., plant and soil metabolism = respiration – photosynthesis; $CO_{2bio}' = CO_{2r} - CO_{2p}$). To isolate the contribution of the terrestrial biosphere to $CO_{2bio}$, we first estimated emissions from biogenic sources, including human/livestock metabolism and biomass burning (Eq. 5).

### 3.2.1 $CO_{2bio}$ from human and livestock metabolism ($CO_{2HLM}$)

Human and livestock metabolism represent a major category of $CO_2$ emissions in urban areas that is often neglected due to its perceived small magnitude compared to fossil fuel emissions (Cai et al., 2022; Wang et al., 2024b). At global and national scales, such emissions to atmosphere are offset by photosynthetic $CO_2$ uptake in agricultural systems, which serve as the primary source of food (directly) or livestock feed (indirectly via meat production). However, in densely urbanized areas like Shenzhen, most of the carbon efflux from human beings was originally taken up as $CO_2$ in surrounding rural regions.

For $CO_{2bio}$ from human and livestock metabolism ($CO_{2HLM}$), we first calculate the ratio of human and livestock metabolic (respiratory plus excretory): fossil emissions using fossil emissions for Shenzhen from emission inventory and derived human and livestock emissions. For human metabolic emissions ($CO_{2HM}$; Table S2), respiratory emission rate were derived from basal metabolic rate (BMR) and physical activity ratio (PAR) for different age-sex groups following the work of Wang et al. (2024b). Excretory emission rates were calculated for different age-sex groups based on an average human mass of 70





kg with an excretory emission rate of 128 g C d$^{-1}$ person$^{-1}$ (Miller et al., 2020; Prairie and Duarte, 2007). Similarly, for livestock metabolic emissions (CO$_{2LM}$; Table S3), respiratory emission rates were derived for different livestock species following the work of Cai et al. (2022). Excretory emission rates were assumed to be zero, as livestock excreta is a significant contributor to CH$_4$ and N$_2$O but not to CO$_2$ (Jun et al., 2002). Human and livestock metabolic CO$_2$ emissions were then calculated as 9.32 Mt (9.38 Mt) and 6.22 kt (3.88 kt), respectively, for 2022 (2023) by multiplying the emission rates with permanent resident population of various age-sex groups or livestock production for different species. Noting that the livestock metabolic emissions are negligible compared with the human metabolic emissions. The ratios of human/livestock emissions to fossil emissions (R$_{HLM}$; Table S4) were then estimated with the fossil emissions from the near-real-time Global Gridded Daily CO$_2$ Emissions Dataset (GRACED) (Liu et al., 2020). It is 0.178 ± 0.031 during April 2022 – April 2023, ranging from 0.119 (January 2023) to 0.233 (June 2022). Applying the correction ratios times CO$_{2ff}$ concentrations (Wang et al., 2024b), we estimated annual CO$_{2HLM}$ concentrations to be 1.3 ± 0.6 ppm, with values ranging from 0.1 ppm to 2.1 ppm (Fig. 4a).

### 3.2.2 CO$_{2bio}$ from biomass burning (CO$_{2BB}$)

CO$_{2bio}$ from biomass burning (CO$_{2BB}$) in Shenzhen cannot be overlooked, as the city is close to Southeast Asia, a region known for high biomass burning emissions (Edgar, 2024). Assuming that CO$_{2BB}$ emissions account for 100 % of CO$_{2bio}$ in EDGAR2024 inventory, we calculate Shenzhen BB:fossil emission ratio (R$_{BB}$; Table S4) by extracted biomass burning and fossil emissions from EDGAR2024 (Edgar, 2024) and GRACED, respectively, and then ratioed. The CO$_{2BB}$ emissions are estimated as 5.05 Mt and 4.93 Mt in 2022 and 2023, respectively. The R$_{BB}$ ratio is calculated 0.092 ± 0.015 during April 2022 – April 2023, ranging from 0.069 (August 2022) to 0.115 (February 2023). Applying the correction ratios multiply by CO$_{2ff}$ concentrations (Wang et al., 2024b; Miller et al., 2020), annual CO$_{2BB}$ concentrations were then estimated as 0.7 ± 0.4 ppm, with values ranging from 0.04 ppm to 1.1 ppm (Fig. 4a).

### 3.2.3 CO$_{2bio}$ from terrestrial biosphere (CO$_{2bio}$')

We calculated the overall biogenic:fossil emission ratios (R$_{bio}$ = R$_{HLM}$ + R$_{BB}$; Table S4) as 0.270 ± 0.035 during April 2022 – April 2023, ranging from 0.203 (January 2023) to 0.325 (June 2022). After applying the overall correction ratios, we isolate the CO$_{2bio}$ contributions from the urban biosphere, define as CO$_{2bio}$' = CO$_{2bio}$ – R$_{bio}$ × CO$_{2ff}$. This adjustment produces an average annual CO$_{2bio}$' enhancement of –3.8 ± 4.8 ppm using Eq. 5, with mean summertime and wintertime CO$_{2bio}$' values of –4.3 ± 6.1 ppm and –2.7 ± 2.2 ppm, respectively (Fig. 4a). On average, individual CO$_{2bio}$' values are 2.0 ppm lower than the uncorrected CO$_{2bio}$ values, with a 7.2% increase in the proportion of negative values.

Based on the above calculations, the three biogenic CO$_2$ components: CO$_2$ from human and livestock metabolism (CO$_{2HLM}$), biomass burning (CO$_{2BB}$), and terrestrial biosphere (CO$_{2bio}$') contribute 17.8 ± 3.1%, 9.2 ± 1.5%, and 73.0 ± 3.5% (relative to fossil fuel emissions), respectively (Table S5). CO$_2$ emissions from human and livestock metabolism (CO$_{2HLM}$) are nearly



double those from biomass burning ($CO_{2BB}$). Shenzhen's HLM contribution (17.8%) exceeds that of Los Angeles (5.7%), Beijing (7.5%), and Paris (15.0%), reflecting its exceptional population density (~9000 $km^{-2}$) driving intense metabolic

fluxes. For biomass burning in Shenzhen (9.2%) versus biofuel use in Los Angeles (10.3%), the closely aligned shares indicate that biomass-related emissions represent a stable source common to these cities. Notably, Shenzhen's terrestrial biosphere contribution (73.0%) is comparable to Los Angeles' long-term maximum (84.0%). This dominance of natural processes in Shenzhen's carbon emissions is attributable to its subtropical climate, high vegetation cover, and rapid soil microbial activity.

We found that the compositional ratios of $CO_{2bio}$ were sensitive to the choice of emission inventory. To quantify this uncertainty, we conducted a sensitivity analysis in the calculation of $CO_{2bio}'$ by testing alternative $R_{bio}$ values of 0.161, 0.215, 0.253, and 0.322 (see Table S5) derived from ODIAC2022 (2021) (Oda and Maksyutov, 2024), MIXv2 (2017) (Li et al., 2024b), EDGARv7.0 (2021) (Crippa et al., 2023), and MEICv1.4 (2020) (Crippa et al., 2023) emission inventories, respectively (Fig. 4b). The annual mean $CO_{2bio}'$, calculated from monthly averages, increases by 0.8, 0.4, 0.1, and –0.4 ppm,

respectively. Despite these adjustments, the seasonal cycle amplitude (defined as the difference between January and September) remains consistent at approximately 11.5 ppm, with a variation of less than 1.4%. This stability is expected, as the correction ratios are applied annually for inventories, and $CO_{2ff}$ does not exhibit significant seasonality. Even for the uncorrected $CO_{2bio}$, the seasonal amplitude of 11.0 ppm is within 4.3% of the amplitude calculated for the two $CO_{2bio}'$ estimates.

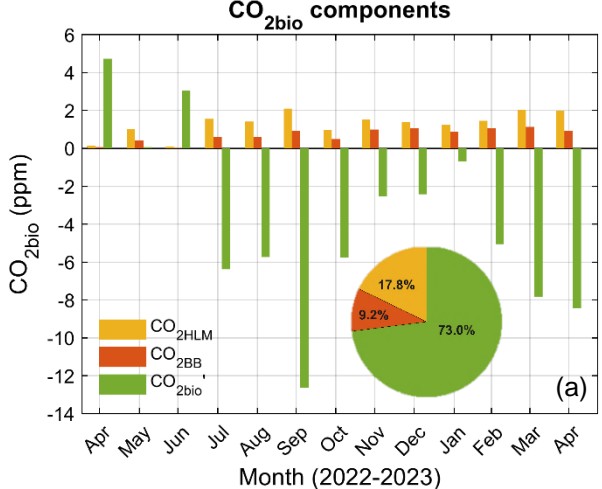

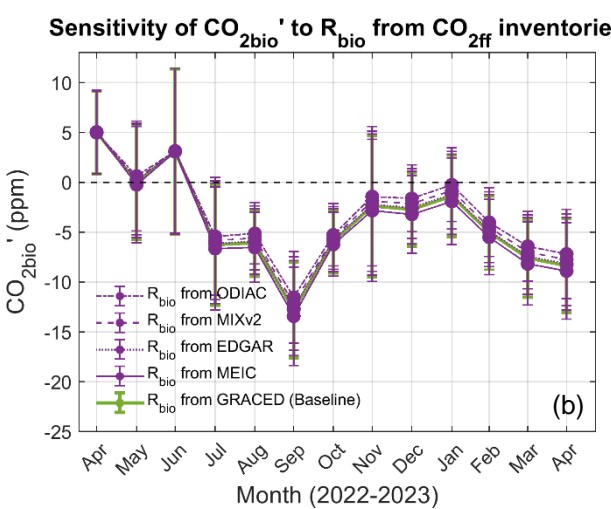




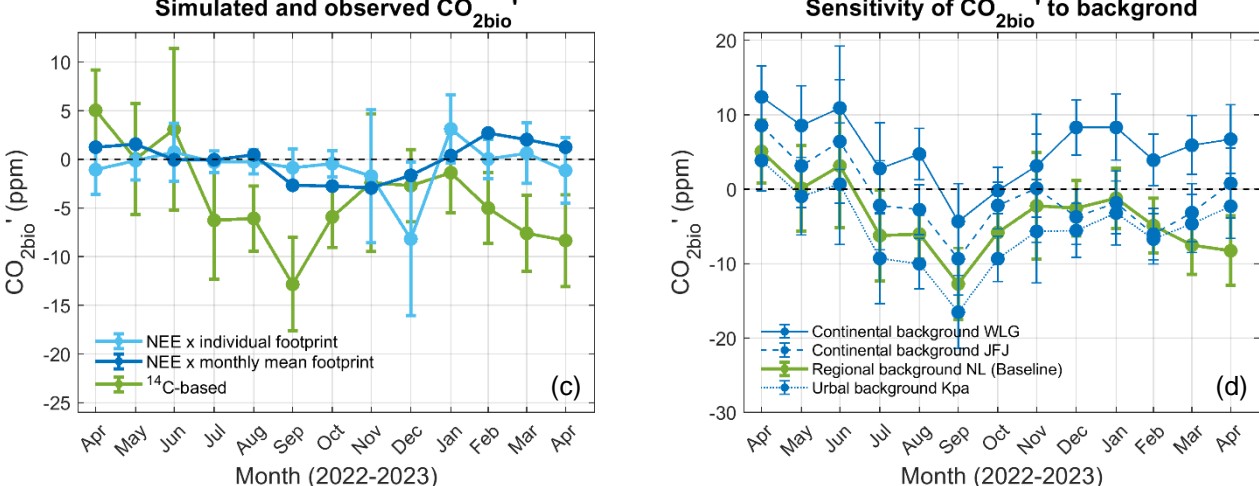

**Figure 4: (a) Monthly concentrations of CO₂bio components: CO₂HLM, CO₂BB and CO₂bio'. (b) Sensitivity of CO₂bio' to biogenic:fossil emission ratios. The green line represents the baseline monthly average CO₂bio' values (0.270 from GRACED); the purple line represents CO₂bio' calculated using the modified biogenic:fossil emission ratios (R_bio) of 0.161, 0.215, 0.253, and 0.322 derived from**
**the ODIAC2022 (2021), MIXv2 (2017), EDGARv7.0 (2021), and MEICv1.4 (2020) inventories, respectively (see Table S5). (c) Comparison of CO₂bio' derived from Shenzhen Δ¹⁴C and CO₂ measurements (green lines) with locally added biospheric CO₂ simulated from the CASA-GFED4s NEE (Randerson et al., 2017; Van Der Werf et al., 2017) using the FLEXPART footprint of the five sampling sites. The simulated CO₂bio' was calculated by convolving the NEE flux field with either individual footprints from the five sites and then averaged (light blue lines) or the sum of the monthly mean footprints for each site (blue lines). (d)**
**Sensitivity of CO₂bio' to backgrounds. The green line represents the baseline monthly average CO₂bio' values; the blue lines represent using WLG (Waliguan), JFJ (Jungfraujoch), and Kpa as the background, respectively, for CO₂ and Δ¹⁴C instead of NL (Nanling).**

## 3.3 Seasonal variations of CO₂bio'

### 3.3.1 Significant seasonal amplitude

We found that the seasonal amplitude of CO₂bio' (CO₂ originating from the urban biosphere) in Shenzhen (11.5 ppm) is approximately 1.5 times the annual mean fossil fuel contribution (7.7 ppm), a significant proportion for a metropolitan area dominated by industrial and transportation emissions. This amplitude in Shenzhen in 2022 is 2.6 times greater than that observed in Los Angeles in 2015 (4.3 ppm), a city with a Mediterranean climate (Miller et al., 2020).

The observed seasonal amplitude of CO₂bio' is also higher than those from simulations. Assuming that the monthly mean NEE from the CASA-GFED4s dataset represents the NEE for all local vegetation cover, we convolved sample footprints at each site with NEE maps, and found the resulting simulated CO₂bio' amplitudes at our measurement sites are smaller than 11.5 ppm and CO₂bio' minima also showed in different months (Fig. 4c). As noted above, sources from urban vegetation must account for the majority of the observed CO₂bio' signal, given that sources like human and livestock metabolism (HLM) and
biomass burning (BB) are purely emissive, while our observations require seasonal carbon sinks. Thus, our results indicate that the CASA-GFED4s dataset may underestimate the seasonal amplitude of NEE fluxes in Shenzhen.



We realized that the amplitude of $CO_{2bio}$' seasonal variations were highly sensitive to background selection. To quantify this effect (Fig. 4d), we analysed $CO_2$ and $\Delta^{14}C$ data from two contrasting backgrounds: Kpa (lacking seasonal cycles) and Waliguan (WLG) at 3,890 m a.s.l. (showing clear seasonality), located ~1,700 km northwest of Nanling (NL) with 2,200 m

elevation difference (Lan et al., 2024; Liu et al., 2024). Despite significant $CO_2/\Delta^{14}C$ differences between NL and these backgrounds, the $CO_{2bio}$' seasonal amplitude remained robust at 12.6–13.3 ppm when using either WLG (seasonal) or Kpa (non-seasonal) as backgrounds. This stability primarily stems from strong local $\Delta^{14}C$ signals at Shenzhen sampling sites. Crucially, NL is the optimal background due to: (i) proximity to urban sites, and (ii) effective capture of $CO_2/\Delta^{14}C$ seasonality, a feature absents at Kpa and geographically distant at WLG (>1,700 km distant).

**3.3.2 Dominant driver factors**

We attempted to investigate the dominant driver factors of the pronounced seasonal amplitude in $CO_{2bio}$'. Firstly, we confirmed that population flow does not contribute significantly to $CO_{2bio}$' seasonal cycle, even though Shenzhen has the largest population flow in China, accounting for 70.8% of its total population. This is supported by the lower $CO_{2bio}$' observed after the Spring Festival (22 January 2023; $2.8 \pm 2.6$ ppm > $0.7 \pm 1.4$ ppm), a period characterized by the highest

population influx, in contrast to the large outflux observed before the festival (Fig. S2a). Moreover, the lack of significant correlation between the monthly average $CO_{2bio}$' and the mobile population further supports this finding ($|r| < 0.48$, $p > 0.39$; Table 1a).

Secondly, we verified that the terrestrial biosphere, mainly plant and soil metabolism (i.e., $CO_2$ released from respiration minus those uptakes from photosynthesis), is the primary contributor to $CO_{2bio}$'. The potential contribution of urban

vegetation to $CO_{2bio}$' is supported by land surface classification derived from the MODIS Vegetation Continuous Fields (Dimiceli, 2024), which indicates an average vegetation cover of 73.3% across the Shenzhen area. Additionally, the footprint-weighted mean vegetation cover in the upwind fetch of our measurement sites ranges from 65.5% to 78.5% (Fig. S3), significantly higher than 14% simulated in Los Angeles (Miller et al., 2020). Analysis of remote sensing and aerial imagery indicates that Shenzhen has a vegetation coverage of 56.2% (with tree coverage of 55.6% and grass coverage of

0.6%) (Qian et al., 2020), while the Shenzhen Statistical Yearbook 2023 reports a greening coverage of 50.8% (Smbs, 2024). Of the city's terrestrial vegetation, artificial evergreen broadleaf forest, garden vegetation, and south subtropical evergreen broadleaf forest collectively account for 85% (Shu et al., 2020). These data highlight the substantial presence of urban vegetation (dominated by trees rather than nontree vegetation; dominated by forest rather than wetland or grassland), supporting its significant potential contribution to $CO_{2bio}$'.

The contribution of urban vegetation to $CO_{2bio}$' is further demonstrated by strong correlations between the monthly mean $CO_{2bio}$' and commonly used vegetation indices (reflecting vegetation growth status), such as the Normalized Difference Vegetation Index (NDVI) and Leaf Area Index (LAI) (Fig. S2b). NDVI, significantly linearly correlated with vegetation distribution density, is widely used to reflect changes in ecological land use (ranged from –1 to 1), with changing positive



values representing changing vegetation coverage fractions. LAI, an important parameter for describing vegetation structure and function, can be used to reflect photosynthesis, water use, and material exchange and energy balance, which significantly influence vegetation-climate feedback mechanisms (Fang et al., 2019). We observed that the monthly averaged $CO_{2bio}'$ shows significant negative correlations with NDVI from both NDVI_China ($r = –0.75$, $p = 0.02$) and GIMMS-3G+ ($r = –0.71$, $p = 0.03$), as well as with LAI from GRDC ($r = –0.82$, $p = 0.007$) (Table 1b). These strong correlations between the monthly average $CO_{2bio}'$ and vegetation indices further support the role of vegetation dynamics in shaping $CO_{2bio}'$ variations.

The negative annual sum of $CO_{2bio}'$ indicates that Shenzhen's terrestrial biosphere serves as a net carbon sink year-round. This finding aligns with observations from the Dinghushan Biosphere Reserve (DBR), a subtropical evergreen mixed forest located 173 km northwest of Shenzhen (Njoroge et al., 2021). Notably, the observed $CO_{2bio}'$ minima occur during July – October, whereas simulated minima shift to September – December (Fig. 4c). A similar delay is reflected in DBR's eddy covariance flux tower data, which show NEE minima during October – January. Crucially, Shenzhen's urban vegetation maintains a winter $CO_2$ sink despite cold and dry conditions (Fig. 4a), consistent with DBR's negative NEE fluxes. It is worth noting that this sink determination is highly sensitive to background selection (Fig. 4d). Both regional and urban background scenarios (NL and Kpa) indicate an annual carbon sink throughout the year. Notably, the biospheric uptake can offset ~63% of fossil fuel emissions during growing seasons.

Thirdly, we identified that the pronounced seasonal amplitude in $CO_{2bio}'$ is primarily driven by atmospheric temperature, rather than precipitation, mediated through time-lagged effects and accumulation effects. Southern China's humid subtropical monsoon climate features a rainy season from June to September (Fig. S2c), typically inducing peak carbon uptake in summer for unmanaged ecosystems. This aligns with the observed $CO_{2bio}'$ minima (indicating peak uptake) from July to October (Fig. 4a), nearly coinciding with Shenzhen's temperature and precipitation maxima (June – September).

The terrestrial biosphere's control on $CO_{2bio}'$ is further evidenced by lagged correlations with climate factors. Observed $CO_{2bio}'$ shows strong negative correlation with temperature (1–2 months lag; $r = –0.80$, $p = 0.01$), weak negative correlation with precipitation (1 month lag; $r = –0.40$, $p = 0.29$), and insignificant correlation with sunshine hours (Table 1c). These patterns mirror known vegetation-climate relationships: temperature directly regulate photosynthesis and respiration, while precipitation affects soil moisture conditions (Piao et al., 2014; Wu et al., 2015; Ding et al., 2020; Tang et al., 2021). These findings suggest that temperature dominates urban vegetation growth and $CO_{2bio}'$ regulation in Shenzhen, overwhelming precipitation effects, which is consistent with broader findings for southern China (Ren et al., 2023; Wang et al., 2024a). Precipitation is no longer a limiting factor for vegetation growth here, as confirmed by Shenzhen's persistently high humidity (FAO/UNEP Aridity Index >1.5 for past nine years; Fig. S4).

The biospheric $CO_2$ seasonality in Shenzhen (monsoon humid subtropical climate) exhibits fundamentally different drivers from Los Angeles (Mediterranean climate) (Miller et al., 2020), as evidenced by their distinct climate-vegetation regimes (Table S6). Shenzhen's dense natural canopy (MODIS VCF: 73%, forest cover: 55.6%) responds strongly to temperature ($r$



= –0.80), yielding a large seasonal amplitude (11.5 ppm) with September minima during monsoon warmth. In contrast, Los Angeles' sparse vegetation (VCF: 14%) depends on irrigated lawns (12% land cover; $r$ = –0.77) for summer carbon uptake, producing smaller fluctuations (4.3 ppm) peaking in July. This divergence stems from their climatic constraints: Shenzhen's abundant rainfall (>1.5 Aridity Index) eliminates water limitations, allowing temperature to dominate phenology, whereas LA's summer drought necessitates artificial irrigation to sustain biospheric activity. Despite comparable populations (~18 million), these mechanisms explain Shenzhen's 2.6-fold greater $CO_{2bio}'$ amplitude and delayed minima, highlighting how climate background governs urban carbon sink functionality.

**Table 1: Correlations between monthly average $CO_{2bio}'$ and (a) population flow, (b) vegetation indices, and (c) climate factors [a] with a lag of –1 to 3 months**

| (a) Population flow [a] | Inflow | Outflow | Net influx |
|---|---|---|---|
| $r$ | –0.02 | –0.20 | 0.47 |

| (b) Vegetation indices [b] | NDVI from NDVI_China | NDVI from GIMMS-3G+ | LAI from GRDC |
|---|---|---|---|
| $r$ | **–0.75\*\*** | **–0.71\*\*** | **–0.82\*\*\*** |

| (c) Climate factors [c] | Temperature (°C) | Precipitation (mm) | Sunshine (hours) |
|---|---|---|---|
| Lag –1 | 0.02 | 0.32 | –0.25 |
| Lag 0 | –0.34 | 0.02 | 0.21 |
| Lag 1 | **–0.80\*\*\*** | –0.40 | –0.15 |
| Lag 2 | **–0.80\*\*\*** | –0.37 | –0.35 |
| Lag 3 | **–0.63\*** | –0.29 | –0.18 |

[a] **Population flow data in Shenzhen is obtained by crawling the migration scale indexes from the Baidu Migration (https://qianxi.baidu.com, last accessed on 20 May 2024) and then using the indexes to divide by a coefficient ($k$). This is based on the assumption that the index is an elementary function mapping the result of the real migrant population. The coefficient $k$ was calculated to be $3.24 \times 10^{-5}$ by combining the Fermat-Euler theorem and parameter estimation with real data (Wang and Yan, 2021). Net influx = inflow – outflow. [b] vegetation indices include Normalized Difference Vegetation Index (NDVI) and Leaf Area Index (LAI), with the former obtained from a daily gap-free NDVI dataset in China (NDVI_China) (Li et al., 2024a; Li, 2024) and the Global Inventory Modelling and Mapping Studies-3rd Generation V1.2 (GIMMS-3G+) dataset (Pinzon et al., 2023), and the latter obtained from the Global Resources Data Cloud (GRDC, www.gis5g.com, last accessed: September 17, 2024). [c] monthly averaged climate data is obtained from the Meteorological Bureau of Shenzhen Municipality (https://weather.sz.gov.cn, last accessed: June 2, 2024).**

## 4 Conclusions and outlook

This study establishes that urban biogenic $CO_2$ constitutes a significant and quantifiable component of megacity carbon budgets. Through atmospheric $\Delta^{14}CO_2$ and $CO_2$ observations with rigorous background selection and bias corrections, we provide precise estimates of $CO_{2bio}$ contributions—essential for avoiding biases in $CO_{2ff}$ estimates derived solely from $CO_2$ measurements. By integrating multi-source data, we quantitatively partition monthly urban $CO_{2bio}$ in Shenzhen into three



distinct components: human/livestock metabolism, biomass burning, and terrestrial biosphere. Our analysis reveals three key insights regarding the terrestrial biosphere component: The urban biosphere functions as a year-round net carbon sink, maintaining negative $CO_{2bio}'$ values even during dry winters; Temperature dominates seasonal dynamics by lagging 1-2 months, driving peak carbon uptake from July to October; High vegetation cover and abundant humidity enable temperature-controlled phenology, generating significant seasonal amplitude. Crucially, this biospheric sink offsets a substantial fraction of concurrent $CO_{2ff}$ emissions during growing seasons—a natural mitigation mechanism unique to subtropical megacities. The predominance of thermal controls over hydrological constraints reveals a distinct bioclimatic regime governing urban carbon cycles.

These findings carry profound global implications: Megacities with extensive vegetation and high humidity—particularly those in tropical and subtropical regions like Kolkata, Dhaka, and Ho Chi Minh City (Dangermond and Meriam, 2022))—likely harbour amplified biospheric signals due to enhanced photosynthesis, potentially yielding larger carbon sink potential. Critically, in such regions, biomass burning emissions can be substantial due to prevalent agricultural residue burning and domestic biofuel use, potentially dominating $CO_{2bio}$ where combustion regulations are weaker. This further complicates emission accounting, as undetected biospheric fluxes may introduce significant biases in top-down $CO_{2ff}$ estimates across diverse climate zones. Such biases have been robustly evidenced by $\Delta^{14}C$ studies from temperate cities, including Indianapolis (Levin et al., 2003), Los Angeles (Miller et al., 2020), Heidelberg (Turnbull et al., 2015), Paris (Lopez et al., 2013), and Krakow (Zimnoch et al., 2012)—confirming that neglecting $CO_{2bio}$, even during winter dormancy, systematically distorts atmospheric $CO_{2ff}$ quantification. Our methodology thus resolves these limitations, providing the necessary framework to evaluate urban greening initiatives like Shenzhen's "Five Year Million Trees" Action Plan (Jiang, 2023), and advances climate-specific carbon accounting essential for evidence-based policymaking.

Limitations primarily concern inventory scalability: As a preliminary investigation into biomass burning contributions, we employed the EDGAR inventory due to its unique incorporates of both open and domestic combustion emissions. However, global-scale inventories like EDGAR may be less suitable for city-level applications (Gurney et al., 2019), as localized policy interventions—Shenzhen's 2017 citywide ban on biomass formed fuel combustion and mandated transition of industrial biomass boilers to clean energy (Sszd, 2017)—can rapidly decouple emissions from socioeconomic trends. Consequently, future work must prioritize developing urban-specific inventories that: (1) distinguish combustion types (open vs. domestic), (2) integrate real-time policy impacts, and (3) quantify sub-city spatiotemporal heterogeneity in biogenic fluxes. Such granular inventories are indispensable for targeted carbon neutrality strategies.

Building on this foundation, while continuous atmospheric $\Delta^{14}CO_2$ campaigns may be impractical for all megacities, periodic $\Delta^{14}C$ monitoring remains indispensable for resolving seasonal to long-term trends in both $CO_{2ff}$ and $CO_{2bio}$. To advance global urban carbon neutrality, we recommend extending observational constraints to diverse climate zones, refining biosphere lag-effect mechanisms, and integrating dynamic vegetation models with high-resolution flux maps—an integrated framework that will significantly enhance urban emissions assessment and mitigation tracking accuracy.



**Code availability**

The FLEXPART 10.4 model is available at https://www.flexpart.eu. Commercial software MATLAB R2023a, public software Python 3.9, and MODIS Reprojection Tool are used for data processing and result visualization.

**Data availability**

The datasets generated in this study are available from the corresponding author on reasonable request. The near-real-time Global Gridded Daily $CO_2$ Emissions Dataset (GRACED) is available at https://www.carbonmonitor-graced.com/index.html.

The Multi-resolution Emission Inventory for China (MEIC) is available at http://meicmodel.org.cn. The Emissions Database for Global Atmospheric Research (EDGAR) Global Greenhouse Gas Emissions are available at https://edgar.jrc.ec.europa.eu. The Open-Data Inventory for Anthropogenic Carbon dioxide (ODIAC) is available at https://db.cger.nies.go.jp/dataset/ODIAC/. The MIXv2 Asian emission inventory (MIXv2) is available at https://csl.noaa.gov/groups/csl4/modeldata/data/Li2023. The Carnegie Ames Stanford Approach Global Fire Emissions

Database Version 4 (CASA-GFED4s) dataset is available at https://daac.ornl.gov/VEGETATION/guides/fire_emissions_v4_R1.html. The CASA-GFED3 dataset is available at http://nacp-files.nacarbon.org/nacp-kawa-01/. The Global Inventory Modelling and Mapping Studies-3rd Generation V1.2 (GIMMS-3G+) NDVI dataset is available at https://doi.org/10.3334/ORNLDAAC/2187. The Moderate Resolution Imaging Spectroradiometer Vegetation Continuous Fields (MODIS VCF, product MOD44Bv061) is available at

https://lpdaac.usgs.gov/products/mod44bv061. The National Centres for Environmental Prediction's Climate Forecast System (CFSv2) Reanalysis data that drive the FLEXPART model is available at https://rda.ucar.edu/datasets/ds094.0.

**Supplement**

The link to the supplementary information available at website.

**Author contributions**

G.Z., Z.Z., Jun Li, and P.L. conceived and designed the study. Nearly all authors contributed to field sampling. Sanyuan Zhu performed accelerator mass spectrometry (AMS) radiocarbon measurements. P.L. and B.L. conducted modeling simulations. P.L. led data processing, inventory analysis, and manuscript drafting. G.Z., Jun Li, Jing Li, Z.C., and Z.Z. provided revisions to the manuscript.



**Competing interests**

The authors declare no competing interest.

**Acknowledgements**

We acknowledge contributions from Professor Zhang's research group (Jiangtao Li, Menghui Li, Yanmin Sun, Kechang Li, Lixin Wang, Yingjian Shao, Ziyang Zhang, Kun He, Chuxin Yao, Run Lin, and Boji Lin) and Mr. Zhou's group (Zhenqiang Huang and Youxin Qiu) for field sampling assistance. Special thanks are extended to Jiangtao Li for sample extraction and

Run Lin for graphite preparation.

**Financial support**

This study was supported by the National Natural Science Foundation of China (NSFC; nos. 42330715, 42103082, and 42203081), China Postdoctoral Science Foundation (Grant no. 2022T150652), Guangdong Provincial Applied Science and Technology Research and Development Program (Grant nos. 2022A1515011271 and 2022A1515011851), Special Research

Assistant Program of the Chinese Academy of Sciences (CAS), and Director's Fund of Guangzhou Institute of Geochemistry, CAS (Grant no. 2021SZJJ-3).

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
