# Peer review of "Constraining urban biogenic CO2 fluxes: Composition, seasonality and drivers from radiocarbon and inventory analysis"

_EGUsphere, 2025_

## Referee Comment (RC1)

**Review of Li et al. (2025), "Constraining urban biogenic CO2 fluxes: Composition, seasonality and drivers from radiocarbon and inventory analysis"**

In their study, Li et al. (2025) use CO2 and  $\Delta^{14}$ CO2 observations from several measurement sites located in Shenzhen, China, to derive the CO2 contribution from the urban biosphere (CO2bio) over the course of one year. In order to separate the CO2 contribution from the terrestrial biosphere alone (CO2bio'), the authors estimate the CO2 contributions from biomass burning and human metabolism using inventory data and footprint modelling. Finally, they discuss the seasonal variation of the CO2bio' signal and its potential drivers.

In my opinion, this is a well-structured study that provides new insights into the biospheric CO2 contributions of a megacity in a humid subtropical climate, and therefore fits well within the scope of ACP. Below are some comments and suggestions that the authors should consider before publication.

**General comment:**

One strength of this study is that the authors perform several sensitivity analyses to assess the robustness of their results. In particular, it's important to investigate the impact of the CO2 and  $\Delta^{14}$ CO2 background on the estimated CO2bio' signals, which can be significant, as demonstrated by the authors. For the background sensitivity analysis, the authors replace the regional (default) background (NL) with a continental (WLG) and an urban (Kpa) background. The latter is derived from an "urban Keeling plot". In my opinion, this urban background is not suitable for this study for two reasons: (1) It seems that the Kpa background is heavily influenced by very few  $\Delta^{14}$ CO2 outliers, which may be affected by nuclear (or oceanic?)  $^{14}$ C contamination. Looking at Fig. 2b, the Kpa background does not appear to be urban at all, as it is even higher than the regional and continental  $\Delta^{14}$ CO2 backgrounds. (2) The Kpa background is constant and exhibits no seasonal variability. Therefore, it is not suitable for investigating the seasonal variations in the CO2bio' contributions, which is one of the authors' main aims.

To create a more reliable urban background, one could consider wind direction and use observations from the upwind urban site as a background for the other urban sites (provided that the upwind site is not affected by very local signals, e.g. during periods of very low wind speeds, which could be excluded). For example, SZ1 could then be used as the background site during westerly winds, while SZ5 could be used during easterly winds. Have the authors attempted something like this? Such a method has already been used in other urban studies (e.g. Lauvaux et al., 2016 for Indianapolis; Staufer et al., 2016 for Paris).

Using an urban background instead of a regional one has the advantage that the resulting CO2bio' signal can be directly attributed to biospheric fluxes between the upwind and downwind sites, i.e. from the city of Shenzhen alone. When a regional background is used, however, the resulting CO2bio' signal may also be influenced by biospheric fluxes from outside Shenzhen.

**Specific comments:**

- p. 1, l. 25: It might be more appropriate to write "fossil contributions" instead of "fossil fluxes".
- p. 1, l. 25-26: Please also mention the year of the study.
- p. 1, l. 28: Do the authors mean "the dominant biogenic *net* CO2 source"? Terrestrial respiration might also be a significant CO2bio source, but it is compensated by photosynthetic uptake.
- p. 1, l. 30: Please indicate which months belong to the "growing season" in this subtropical climate.
- p. 3, l. 80: Maybe "significant" is a too strong word here, given that the reported biospheric sink (NEE) is less than 1% of the reported fossil CO2 emissions (if I got it right).
- p. 6, l. 147: When and how often do such situations with "obvious low  $\Delta$ 14C outliers" occur?
- Fig. 2: While the continental WLG background shows similar CO2 concentrations to the continental JFJ background site, it's surprising to see that WLG shows much lower  $\Delta^{14}$ CO2 values. In fact, the WLG  $\Delta^{14}$ CO2 values are even lower than those from the regional NL background. Do the authors have an explanation for this?
- p. 7, l. 166: Please explain the acronyms "CO2r" and "CO2p".
- p. 7, l. 168: Why can the authors ignore contributions from air-sea exchange, given that marine ecosystems cover 36% of the Shenzhen target region (see I. 82 in the manuscript)? Is the  $\Delta^{14}$ C signature of the oceanic CO2 fluxes similar to that of the atmosphere? It would be helpful to include a brief explanation of this here.
- p. 7, l. 191-193: How big is the model domain? Is the NL background site representative of the boundary of the model domain? I think this is important to know when comparing the simulated and observed CO2bio' in Fig. 4a.
- p. 8, l. 225: Please specify which months belong to "winter".
- p. 9, l. 227: Why are some negative CO2ff values expected? In June, almost all CO2ff estimates are negative, which can hardly be explained by  $\Delta^{14}$ CO2 measurement uncertainties alone. Do the authors have an explanation for this? Is the  $\Delta^{14}$ CO2 background not suitable at that time, e.g., were there easterly winds at that time, meaning that the NL background site was downwind of the Shenzhen target area and received polluted air? Or could these high  $\Delta^{14}$ CO2 values be explained by  $^{14}$ C contamination (e.g. from nuclear emissions) at the Shenzhen sites. I think it is important to discuss this time period in more detail, as it is the only time when there are positive CO2bio' contributions (in Fig. 4b).

- p. 9, l. 231-232: The authors mention that CO2ff concentrations in Shenzhen are relatively low compared to other cities, such as Paris and Los Angeles. However, this might depend heavily on the chosen  $\Delta^{14}$ CO2 background, as well as on atmospheric mixing within the boundary layer. For example, for the CO2ff estimation in Paris (Lopez et al., 2013), Mace Head (MHD) was used as the  $\Delta^{14}$ CO2 background, which represents clean Atlantic air. In contrast, the regional NL background used in this study shows significant  $\Delta^{14}$ C depletion (e.g. compared to JFJ, see Fig. 2b), especially in winter. Could this explain the smaller CO2ff values in Shenzhen compared to other cities such as Paris?
- Fig. 3: The pluses representing the NL data in panels (a) and (b) are missing.
- Sect. 3.2: Section 3.2 includes parts of methodological descriptions in each subsection. Moving these to the Methods section would make Sect. 3.2 more concise and focused on the results.
- p. 10, l. 262-264: This sentence is difficult to understand. I would suggest rephrasing it.
- p. 11, l. 296-297: I'm not sure how these values have been calculated. For example, in the case of CO2bio', is the  $73.0 \pm 3.5\%$  the (annual) average of all CO2bio'/CO2ff sample ratios? If so, you might want to write "relative to fossil fuel *contributions*" in line 297. Moreover, as a seasonal cycle is expected, it would be interesting to have the values of the terrestrial biosphere for winter and summer separately.
- p. 12, l. 313-314: Which "two" CO2bio' estimates are the authors referring to? If I understand correctly, CO2bio' estimates were calculated from a total of *five* different emission inventories.
- p. 13, l. 338-341: How large is the impact of the vegetation outside of Shenzhen on the simulated and observed CO2bio' estimates? Could the authors quantify how much of the CO2bio' signal comes from outside Shenzhen using the FLEXPART footprints?
- p. 14, l. 345-347: However, the *absolute* CO2bio' contributions change a lot. I think it is worth discussing this a bit more. Although the authors focus on seasonality here, I think the reader would benefit from such a discussion to better understand the differences caused by the various backgrounds. For example, the WLG background would turn the vegetation into a net CO2bio' source (see Fig. 4d)!
- p. 14, l. 355-356: Why did the authors calculate the correlation between CO2bio' and the mobile population? As I understand it, population flow may impact CO2HLM (via respiration) and CO2ff (via use of fossil fuels), but I do not understand how it could impact CO2bio' directly. Please clarify.
- p. 15, l. 387-388: How sensitive is this estimate to the choice of the CO2 and  $\Delta^{14}$ CO2 background?
- Tab. 1: Please explain the stars (\*) and the blue highlighting color in Table 1.

p. 17, l. 446: Please amend the references: The study by Levin et al. (2003) was done in Heidelberg, and the study by Turnbull et al. (2015) was in Indianapolis.

p. 17, l. 459-463: Here, one should also emphasize the importance of having suitable CO2 and  $\Delta^{14}$ CO2 background observations, given their substantial impact on the CO2bio' estimates.

**Technical corrections:**

```
p. 5, l. 132: "winter" -> "western"
```

p. 11, l. 285-286: Do you mean: "By multiplying the correction ratios by the CO2ff concentrations, annual CO2BB concentrations were then estimated as ... "

```
Fig. 4: "Urbal" -> "Urban" in the legend of Fig. 4d
```

**References:**

Lauvaux, T., et al. (2016), High-resolution atmospheric inversion of urban CO2 emissions during the dormant season of the Indianapolis Flux Experiment (INFLUX), J. Geophys. Res. Atmos., 121, 5213–5236, doi:10.1002/2015JD024473.

Staufer, J., Broquet, G., Bréon, F.-M., Puygrenier, V., Chevallier, F., Xueref-Rémy, I., Dieudonné, E., Lopez, M., Schmidt, M., Ramonet, M., Perrussel, O., Lac, C., Wu, L., and Ciais, P.: The first 1-year-long estimate of the Paris region fossil fuel CO2 emissions based on atmospheric inversion, Atmos. Chem. Phys., 16, 14703–14726, https://doi.org/10.5194/acp-16-14703-2016, 2016.